# High-pressure phase diagrams of FeSe$_{1-x}$Te$_x$: correlation between suppressed nematicity and enhanced superconductivity

K. Mukasa[1], K. Matsuura[1], M. Qiu[1], M. Saito[1], Y. Sugimura[1], K. Ishida [1], M. Otani[2], Y. Onishi [2], Y. Mizukami[1,2], K. Hashimoto [1,2], J. Gouchi[3], R. Kumai [4], Y. Uwatoko [3] & T. Shibauchi [1,2✉]

The interplay among magnetism, electronic nematicity, and superconductivity is the key issue in strongly correlated materials including iron-based, cuprate, and heavy-fermion super-conductors. Magnetic fluctuations have been widely discussed as a pairing mechanism of unconventional superconductivity, but recent theory predicts that quantum fluctuations of nematic order may also promote high-temperature superconductivity. This has been studied in FeSe$_{1-x}$S$_x$ superconductors exhibiting nonmagnetic nematic and pressure-induced anti-ferromagnetic orders, but its abrupt suppression of superconductivity at the nematic end point leaves the nematic-fluctuation driven superconductivity unconfirmed. Here we report on systematic studies of high-pressure phase diagrams up to 8 GPa in high-quality single crystals of FeSe$_{1-x}$Te$_x$. When Te composition $x$(Te) becomes larger than 0.1, the high-pressure magnetic order disappears, whereas the pressure-induced superconducting dome near the nematic end point is continuously found up to $x$(Te) $\approx$ 0.5. In contrast to FeSe$_{1-x}$S$_x$, enhanced superconductivity in FeSe$_{1-x}$Te$_x$ does not correlate with magnetism but with the suppression of nematicity, highlighting the paramount role of nonmagnetic nematic fluc-tuations for high-temperature superconductivity in this system.

[1] Department of Advanced Materials Science, University of Tokyo, Kashiwa, Chiba 277-8561, Japan. [2] Department of Applied Physics, University of Tokyo, Hongo, Tokyo 113-8656, Japan. [3] Institute for Solid State Physics, University of Tokyo, Kashiwa, Chiba 277-8581, Japan. [4] Condensed Matter Research Center and Photon Factory, IMSS, KEK, Tsukuba, Ibaraki 305-0801, Japan. ✉email: shibauchi@k.u-tokyo.ac.jp

Electronic nematic states, which break rotational symmetry of the underlying lattice, often emerge in strongly correlated electron systems[1] including cuprate superconductors[2,3] and heavy-fermion compounds[4,5]. The most dramatic examples can be found in iron-based superconductors[6], where a clear structural phase transition at $T_s$ from high-temperature tetragonal to low-temperature orthorhombic phase is driven by an electronic nematic order, whose origin and relation to high-temperature superconductivity have been longstanding issues[7]. Theoretical studies have found that the nematic quantum fluctuations, which are expected to be enhanced around the end point of an electronic nematic phase, can mediate Cooper pairing[8–10]. This mechanism of unconventional superconductivity is distinctly different from the one based on spin fluctuations[11,12], and its experimental verification remains elusive. This is partly due to the closeness between nematic and antiferromagnetic orders in iron-pnictide superconductors, and the enhanced superconductivity can be found near both ends of these two ordered phases, where both magnetic and nematic fluctuations are enhanced.

FeSe with a superconducting transition temperature $T_c \approx 9$ K serves as an ideal platform to study the relationship between the nematicity and superconductivity, because unlike other iron-based superconductors its nematic order below the structural transition at $T_s \approx 90$ K is accompanied by no magnetic order[13–16]. Especially, recent success of high-quality single-crystal growth by the chemical vapor transport technique[17] has opened a pathway to study intrinsic physics in this system. The nematic order in FeSe can be completely suppressed by isovalent S substitution for Se site without inducing magnetic order[18], whereas antiferromagnetism can be induced by the application of hydrostatic pressure[19–21]. The temperature ($T$) versus pressure ($P$) phase diagrams have been studied in vapor-grown $FeSe_{1−x}S_x$ crystals[22], which reveals that $T_c$ can be enhanced above 30 K near the ends of the pressure-induced magnetic phase but $T_c$ stays low where the nematic phase vanishes. Although the pressure-induced antiferromagnetism accompanies the orthorhombic structure and thus it also has nematicity[22,23], a direct link between nematic fluctuations and enhanced superconductivity has not been found in $FeSe_{1−x}S_x$.

Recent studies of quasiparticle excitations in the superconducting state of $FeSe_{1−x}S_x$ have revealed that there is an abrupt change in the superconducting properties on the verge of nematic quantum phase transition at S composition $x(S) \approx 0.17$, above which significant low-energy density of states of quasiparticles suddenly appears[24,25]. This implies that the two superconducting states in the nematic and tetragonal phases are fundamentally different. Indeed, recent theory suggests that a very exotic superconducting state having Bogoliubov–Fermi surface may appear in the tetragonal side of $FeSe_{1−x}S_x$[26]. Thus, the absence of enhanced $T_c$ near the nematic end point does not immediately rule out the important role of nematic fluctuations in this system. It has also been suggested that nematic fluctuations could be quenched by the strong coupling to the lattice or local strain effects in $FeSe_{1−x}S_x$ from quantum oscillation studies showing the absence of mass divergence near the nematic end point[27], although the non-Fermi liquid behaviors are found in transport properties[28,29]. This situation calls for a different system to study the relationship between nematicity and superconductivity. Here, we focus on $FeSe_{1−x}Te_x$, in which isovalent substitution of larger Te ions corresponds to negative chemical pressure in contrast to positive chemical pressure in $FeSe_{1−x}S_x$. From the detailed $T$–$P$ phase diagrams over a wide $x(Te)$ range, we find a correlation between the suppression of nonmagnetic nematicity and enhanced superconductivity, which supports the unconventional superconductivity promoted by nematic fluctuations in this system.

## Results

**Temperature-substitution phase diagram of $FeSe_{1−x}Te_x$.** To study the phase diagrams, it is essentially important to use high-quality single crystals. It has been known for $FeSe_{1−x}Te_x$ that phase separation occurs in the region of $0.1 \lesssim x(Te) \lesssim 0.4$ for bulk crystals[30]. Owing to recent efforts on the crystal growth, single crystals of $FeSe_{1−x}Te_x$ have been obtained for $0 \le x(Te) \lesssim 0.41$ by a flux method with temperature gradient conditions, covering a few compositions inside the phase separation region[31]. Here, we use the chemical vapor transport technique (see "Methods" section), which has been successful for obtaining high-quality single crystals of $FeSe_{1−x}S_x$ up to $x(S) \sim 0.25$ without excess Fe ions. We are able to obtain a series of single crystals of $FeSe_{1−x}Te_x$ up to $x(Te) \sim 0.5$. Figure 1a–c shows the results of X-ray diffraction (XRD) analysis at room temperature. The lattice constants $a$ and $c$, as well as the chalcogen height from the Fe plane, change linearly with Te composition $x(Te)$ within the experimental error, showing that Vegard's law holds with no phase separation.

The temperature dependence of in-plane resistivity $\rho$ normalized at the 200 K value is shown in Fig. 1d. All samples for $0 \le x(Te) \lesssim 0.50$ exhibit metallic behavior with a clear kink anomaly of $\rho(T)$ at the nematic transition temperature $T_s$ up to $x(Te) \approx 0.48$ (Supplementary Fig. 1), which would be smeared out in dirty crystals with excess Fe ions[32]. These results indicate that our vapor-grown crystals are of high quality. The nematic transition is also checked by the low-temperature synchrotron XRD (Fig. 1e, f), which clearly shows the splitting of Bragg peaks indicative of the tetragonal to orthorhombic structural transition. As $x(Te)$ increases, the nematic transition temperature is lowered, and at the same time the orthorhombicity $\delta = (a_o − b_o)/(a_o + b_o)$ is systematically suppressed.

Figure 2a shows the temperature-substitution phase diagram obtained from our resistivity and XRD measurements. The nematic transition temperature $T_s$ decreases almost linearly with $x(Te)$ and is completely suppressed at around $x(Te) \approx 0.50$. The superconducting transition temperature $T_c$ first decreases and has a minimum at around $x(Te) \approx 0.30$, and then turns to increase. Across the nematic transition line, $T_c$ continues to increase and reaches 12.7 K at $x(Te) \approx 0.50$ (Fig. 2b), which is close to the optimum $T_c \approx 14$ K ($x(Te) \approx 0.6$) in this system[33]. This nonmonotonic $T_c(x)$ behavior is consistent with the previous report[31], although the $x(Te)$ value at which the minimum appears is slightly different ($x_{min} \approx 0.19$). It has been argued that the minimum in $T_c$ may be attributed to the effect of sample disorder because the residual resistivity ratio of the sample $x_{min} \approx 0.19$ studied in ref. [31] is relatively small. In our systematic study with much more data points, however, $\rho(200\,K)/\rho(15\,K)$ decreases monotonously with $x(Te)$ (Fig. 2c). This indicates that the increase of $T_c$ above $x(Te) \approx 0.30$ has an intrinsic origin. In other words, there must be some mechanism that enhances $T_c$ toward the high concentration side. As discussed later, nematic fluctuations that are expected to be the largest near the end point of nematic order ($x(Te) \approx 0.50$) can promote such an enhancement of $T_c$.

**Temperature–pressure phase diagrams.** Having established the $T$–$x(Te)$ phase diagram of $FeSe_{1−x}Te_x$, we now investigate the hydrostatic pressure effect. In Fig. 3a–h, we show the evolution of the resistivity curve $\rho(T)$ under pressure with increasing Te composition, measured by using a constant-loading cubic anvil cell (CAC) (see "Methods" section). With applying pressure, the nematic transition at $T_s$ observed at ambient pressure for $x(Te) < 0.50$ is suppressed and disappears at $P \lesssim 2$ GPa. In $x(Te) \approx 0.04$ sample, the $\rho(T)$ curve exhibits a clear upturn at 2 GPa and a kink at $3 \le P \le 5$ GPa (Fig. 3a). It has been shown that similar upturn and kink behaviors are observed in FeSe under pressure at

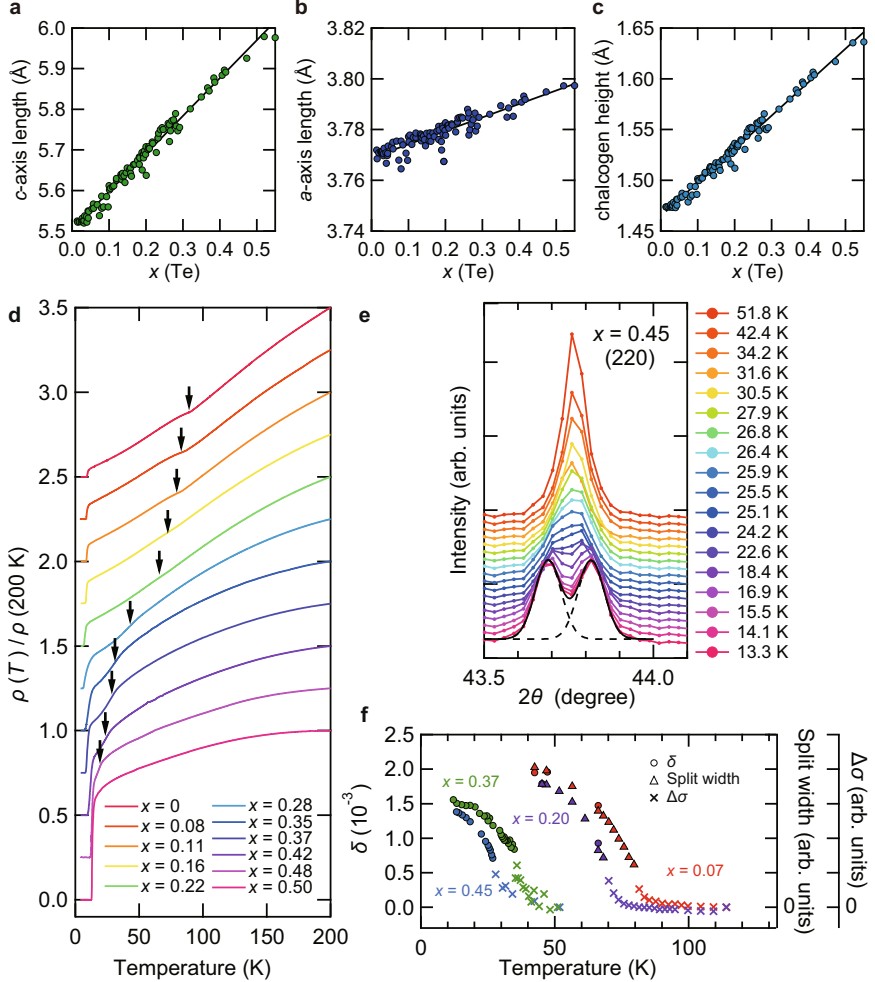

**Fig. 1 Evolution of structural parameters and resistivity with Te composition in single crystals of FeSe$_{1-x}$Te$_x$ at ambient pressure. a–c** Lattice parameters and Te composition $x$(Te) determined by X-ray diffraction (XRD). Lattice constants $c$ (**a**) and $a$ (**b**) as well as the chalcogen height from the Fe plane (**c**) are shown as a function of $x$(Te). Solid line represents the linear $x$(Te) dependence (Vegard's law). **d** Temperature dependence of the in-plane resistivity $\rho$ normalized by the value at 200 K for $0 \leq x$(Te) $\lesssim 0.50$. Each curve is shifted vertically for clarity. The resistive anomalies associated with the nematic transition temperature $T_s$ (black arrows) are determined by the minimum ($0 \leq x$(Te) $\lesssim 0.22$) or the maximum ($0.28 \lesssim x$(Te) $\lesssim 0.50$) of the derivative curves $d\rho(T)/dT$. **e** XRD intensity as a function of the scattering angle $2\theta$ near the (220) Bragg peak measured at several temperatures for $x \approx 0.45$. Each curve is shifted vertically for clarity. The black solid and dashed lines indicate the result of the two-peak fitting for the data at 13.3 K. **f** The orthorhombicity $\delta = (a_o - b_o)/(a_o + b_o)$ estimated from the two-peak fitting of ($hk0$) Bragg peaks for $x$(Te) $\approx 0.07, 0.20, 0.37,$ and 0.45 (circles, left axis) and the split width of ($hkl$) Bragg peaks for $x$(Te) $\approx 0.07$ and 0.20 (triangles, right axis) as a function of temperature. We also plot additional width $\Delta\sigma$ of single peak near the transition (crosses, right axis), estimated by the standard deviation $\sigma$ of the single Gaussian fitting subtracted by the value at the maximum temperature measured for each sample.

the magnetic transition temperature $T_m$[21], below which stripe-type antiferromagnetic order similar to that found in other iron-pnictide superconductors sets in[23,34,35]. Here, the competition between the decrease in carrier concentrations and the decrease in scattering rate by the antiferromagnetism results in either upward or downward change in the resistivity depending on the slight change in the condition. In fact, it has been seen in FeSe that, by application of magnetic field, the downward kink behavior of $\rho(T)$ below $T_m$ gradually changes to the upward jump[21,36], similar to the change between the present 2 and 3 GPa data. We thus follow the procedure of ref. [21] to determine the magnetic transition temperatures ($T_m$) by using a maximum or minimum in $d\rho/dT$ (Supplementary Figs. 2–5). The superconducting critical temperature $T_c$ is determined by the zero resistivity. The obtained temperature–pressure phase diagram is shown in Fig. 4a.

The resistivity anomalies associated with the pressure-induced magnetism can be seen up to $x$(Te) $\approx 0.10$ (Fig. 3b, c). As seen in the

phase diagrams in Fig. 4b, c, the pressure range in which the magnetic phase appears is extended to $1 \lesssim P \lesssim 5$ GPa for $x$(Te) $\approx 0.06$, while for $x$(Te) $\approx 0.10$ the magnetic phase appears in two separated pressure regions; around 1 and $5 \lesssim P \lesssim 7$ GPa. Similar features have also been reported in the pressure phase diagrams of FeSe$_{1-x}$S$_x$ ($0 \leq x$(S) $\lesssim 0.17$), where the magnetic phase moves to higher pressure range as $x$(S) increases[22] and the magnetic phase is observed inside the nematic phase for $x$(S) $\lesssim 0.10$[37]. For $x$(Te) $\gtrsim 0.14$, we cannot find any anomalies associated with the magnetic transition in the measurement range up to 8 GPa (Fig. 3d–h). Consequently, only the nematic and the superconducting phases exist in the temperature–pressure phase diagrams for $0.14 \lesssim x$(Te) $\lesssim 0.38$ as shown in Fig. 4d–g. This is in stark contrast to the case of FeSe$_{1-x}$S$_x$ in which the dome-shaped magnetic phase centered at $P \sim 5$ GPa persists at least up to $x$(S) $\approx 0.17$ where the nematic phase disappears[22]. Remarkably, the superconducting phase in $T$–$P$ diagrams continues to exhibit a dome shape in a wide range of $x$

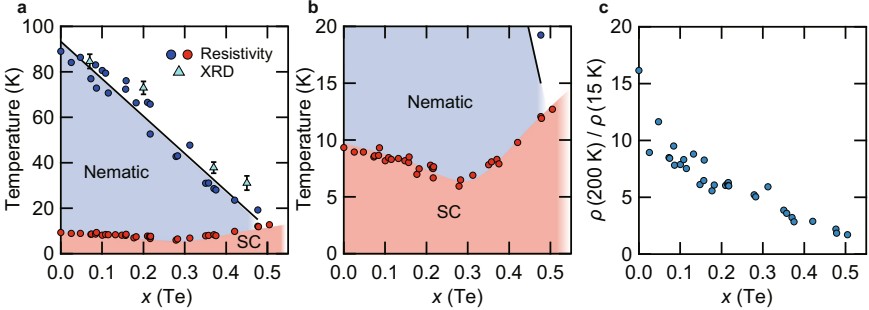

**Fig. 2 Temperature versus Te composition phase diagram of FeSe$_{1-x}$Te$_x$ at ambient pressure. a** Nematic and superconducting transition temperatures as a function of $x$(Te). The blue and red circles represent the nematic ($T_s$) and superconducting ($T_c$) transition temperatures, respectively, determined by the resistivity measurements. The light blue triangles represent $T_s$, determined by the splitting of the Bragg peaks in the XRD measurements. The black line is a least squares $x$(Te)-linear fit to the $T_s$ data from the resistivity measurements. The color shades for the nematic and superconducting (SC) states are the guides to the eyes. Error bars represent the uncertainty in determining $T_s$ from the data in Fig. 1f. **b** The same as in **a**, but the temperature range is 0–20 K. **c** Dependence of $\rho(200\,\text{K})/\rho(15\,\text{K})$ on $x$(Te) extracted from the resistivity data.

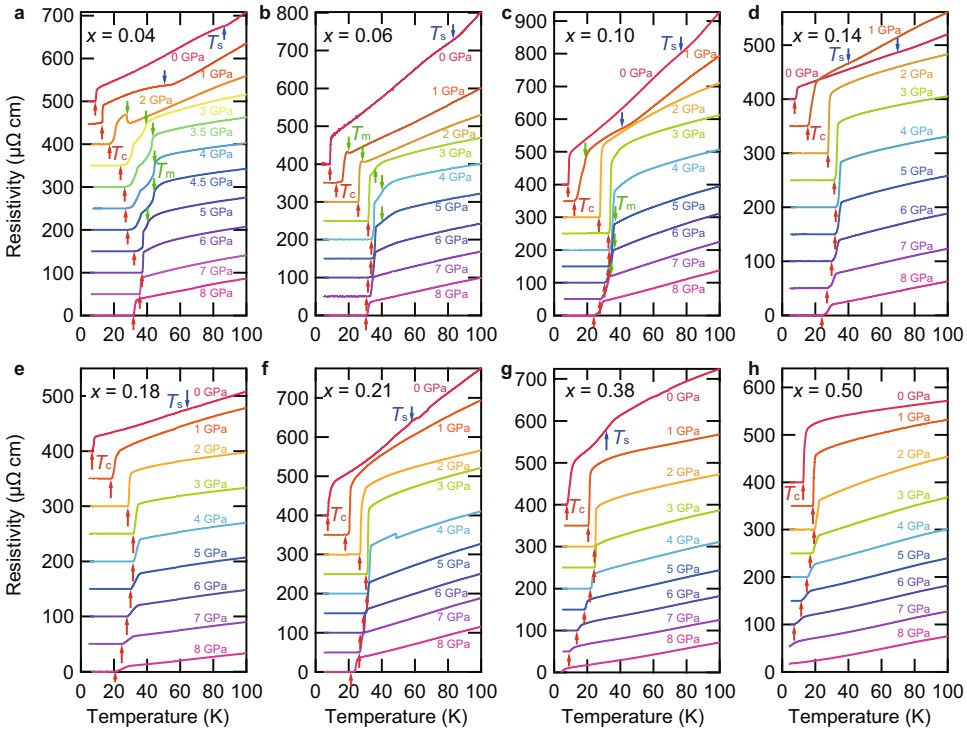

**Fig. 3 Evolution of the temperature dependence of resistivity with pressure.** Temperature dependence of resistivity in FeSe$_{1-x}$Te$_x$ below 100 K at different pressures up to 8 GPa for $x$(Te) ≈ 0.04 (**a**), 0.06 (**b**), 0.10 (**c**), 0.14 (**d**), 0.18 (**e**), 0.21 (**f**), 0.38 (**g**), and 0.50 (**h**). The data are vertically shifted for clarity. The resistive anomalies at transition temperatures $T_s$ (blue), $T_m$ (green), and $T_c$ (red) are indicated by arrows.

(Te) even after the disappearance of the magnetic phase, although the maximum $T_c$ decreases to ~20 K in $x$(Te) ≈ 0.50 sample (Fig. 4h). For $x$(Te) ≈ 0.50, the initial increase of $T_c$ at low pressures is consistent with the previous study[38].

## Discussion

In the case of FeSe$_{1-x}$S$_x$, the high-$T_c$ superconducting phase always locates near the ends of the pressure-induced dome-shaped magnetic phase, implying the intimate relation between antiferromagnetism and high-$T_c$ superconductivity[22]. However, this is not the case in FeSe$_{1-x}$Te$_x$. For $0.14 \lesssim x$(Te) < 0.50, the pressure phase diagrams show superconducting domes at the high-pressure side of the nematic phase and no magnetic phase is found up to 8 GPa (Fig. 4d–g). The difference between S and Te substitutions can also be seen clearly in the $T$–$P$–$x$ three-dimensional phase

diagram in Fig. 4i, combining S and Te substitutions corresponding to positive and negative chemical pressure, respectively. The magnetic phase in FeSe$_{1-x}$Te$_x$ disappears for $x$(Te) > 0.10 where the nematic phase still exists, and the superconducting dome continues to the high Te composition side without magnetism. This shows a clear contrast to the S substitution case, where the magnetic dome stays around ~5 GPa even at the highest composition of $x$(S) = 0.17, and $T_c$ is enhanced near both ends of the magnetic dome. The nematic order at low pressure region has also significant asymmetry between positive and negative chemical pressure: in FeSe$_{1-x}$S$_x$, $T_s$ vanishes at $x$(S) ≈ 0.17 above which $T_c$ is reduced abruptly[24,25], but in FeSe$_{1-x}$Te$_x$, $T_s$ persists up to $x$(Te) ≈ 0.5 above which $T_c$ continues to increase (Fig. 2b).

The new phase diagrams of FeSe$_{1-x}$Te$_x$ indicate that the enhanced superconductivity correlates with the suppression of

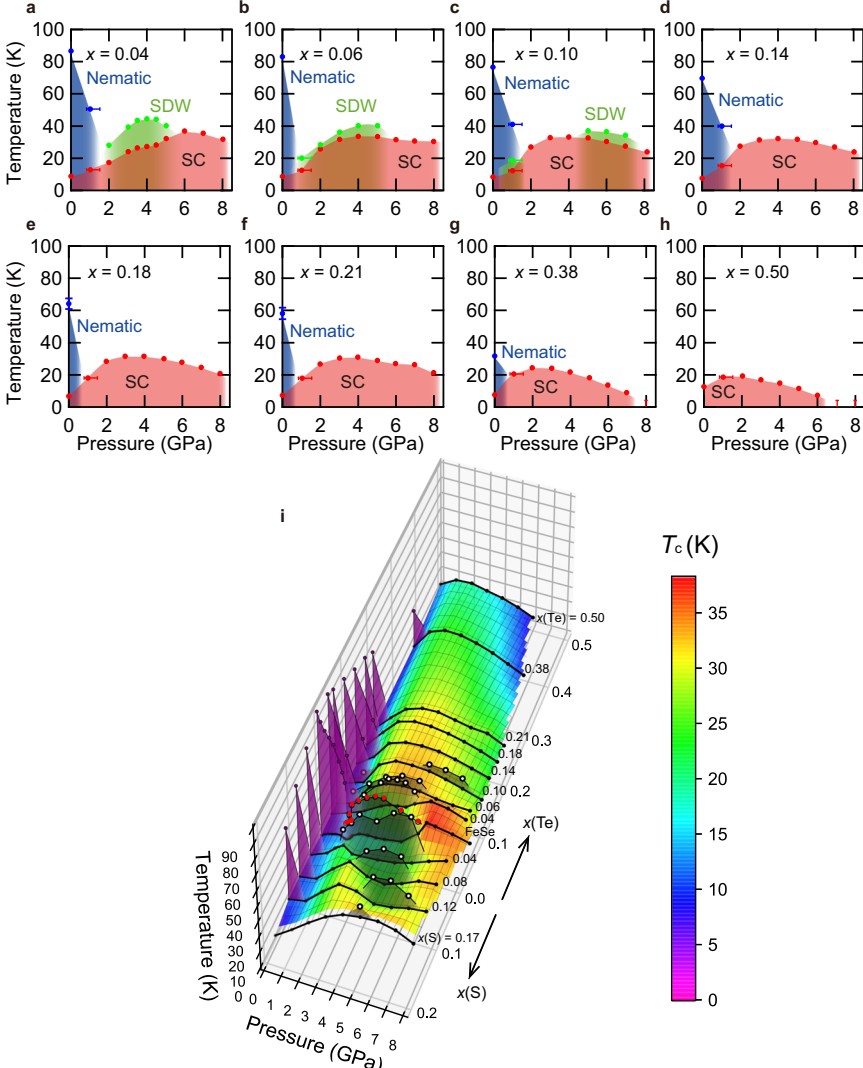

**Fig. 4 Temperature–pressure phase diagrams in FeSe$_{1-x}$Te$_x$.** Pressure dependence of $T_c$, $T_s$, and $T_m$ indicated by red, blue, and green circles, respectively, for $x$(Te) ≈ 0.04 (**a**), 0.06 (**b**), 0.10 (**c**), 0.14 (**d**), 0.18 (**e**), 0.21 (**f**), 0.38 (**g**), and 0.50 (**h**). The color shades for the nematic, spin density wave (SDW), and superconducting (SC) states are the guides to the eyes. The error of pressure for $P < 2$ GPa is relatively large (see error bars for 1 GPa) compared to higher pressures. The errors of $T_s$ are estimated from the least squares fit in Fig. 2a. **i** Three-dimensional electronic phase diagram, temperature versus pressure and Te concentration $x$(Te), of FeSe$_{1-x}$Te$_x$, combined with the reported $T$–$P$–$x$(S) phase diagram of FeSe$_{1-x}$S$_x$ ($0 \leq x$(S) $\lesssim 0.17$)[22]. The surface plot shows $T_c$ and the purple and white circles represent $T_s$ and $T_m$, respectively. The red circles represent $T_m$ of FeSe. The gray and purple shadowed areas indicate the magnetic and nematic phases, respectively.

nematic phase, not with the magnetism. This is consistent with the NMR measurements suggesting that FeSe$_{0.42}$Te$_{0.58}$ does not have any significant antiferromagnetic spin fluctuations[39] with ($\pi$, $\pi$) wave vector determined by the neutron scattering measurements[40,41]. We note that the maximum $T_c$ is attained at somewhat different point from the extrapolated nematic end point. This may be related to the fact that even when the enhanced quantum fluctuations near the critical point enhance the pairing interactions, the effect of quasiparticle damping may also become significant, which could suppress actual $T_c$ at the critical point. Indeed, in the theory of the ferromagnetic spin-fluctuation-based unconventional superconductivity[42], $T_c$ is suppressed just at the critical point but becomes highest not far from the critical point. Therefore, our results in FeSe$_{1-x}$Te$_x$, which reveal the superconducting dome with a broad peak not far from the nematic end point, support the idea that the quantum fluctuations of nonmagnetic nematic ordered phase can promote superconductivity in this system[8–10].

In the pressure phase diagram of FeSe, the suppression of $T_c$ is found inside the pressure-induced magnetic phase, showing a kink behavior of $T_c(P)$ at the crossing point with $T_m(P)$[21], which can be explained by the competition mechanism between magnetism and superconductivity. The competition between nematicity and superconductivity can also explain the opposite trends between $T_c$ and $T_s$ as functions of $x$(Te) and pressure inside the nematic phase. However, this competition alone cannot explain the superconducting dome we observed centered outside the nematic phase. An important point is that the superconducting domes are found near the nematic end point, not close to the magnetic phase, which implies a close relationship between nematic fluctuations and enhanced superconductivity. This does not contradict the competition effect inside the ordered phase, because nematic fluctuations are expected to be suppressed with the development of nematic order.

An obvious question is why this correlation between superconductivity and nematicity is not seen in FeSe$_{1-x}$S$_x$. One possibility

is that in the tetragonal (nonnematic) phase of $FeSe_{1-x}S_x$, an exotic superconducting state emerges with relatively low $T_c$, which is distinctly different from the superconducting states of other FeSe-based materials. Recent specific heat and scanning tunneling spectroscopy measurements have found that the tetragonal $FeSe_{1-x}S_x$ samples exhibit anomalously large low-energy quasiparticle excitations in the superconducting state[24,25]. For example, the zero-bias conductance in the tunneling spectra as a function of $x$(S) jumps at the nematic end point from essentially zero to a fraction of the normal-state value. Such a superconducting state with substantial low-energy quasiparticle density of states can be consistently explained by the presence of Bogoliubov–Fermi surface, which has been recently suggested theoretically[26]. Although further studies are needed to clarify the microscopic mechanism of such an exotic superconducting state, this suggests that $FeSe_{1-x}S_x$ may not be a suitable system to use the phase diagram and the $x$(S) dependence of $T_c$ to discuss the impact of nematic fluctuations on superconductivity.

In contrast, the fully gapped superconductivity is found in the tetragonal phase of $FeSe_{1-x}Te_x$[43], which rules out the exotic state with Bogoliubov–Fermi surface in this system. Thus, our observation of the enhanced $T_c$ correlated with the suppression of nematicity in this nonmagnetic system implies that the nematic fluctuations play a significant role for high-temperature superconductivity.

In summary, by using high-quality vapor-grown single crystals of $FeSe_{1-x}Te_x$ ($0 \le x \le 0.5$), we have established temperature–pressure–composition phase diagrams in a wide range of pressure up to 8 GPa. The superconducting dome close to the nematic phase with no magnetism is observed, implying that the nematic fluctuations can promote high-$T_c$ superconductivity in this system.

## Methods

**Single crystals**. Single crystals of $FeSe_{1-x}Te_x$ ($0 \le x$(Te) $\lesssim 0.55$) have been grown by the chemical vapor transport technique. Fe, Se, and Te powders were mixed and sealed in a quartz ampoule with $AlCl_3$ and KCl as transport agents[17,44]. The atomic ratio of Fe to Se and Te was 1.1:1 and the total mass of the starting materials and the transport agents were 1.05 and 2.45 g, respectively. The growth time was 1–2 weeks. The temperatures of the source and sink sides were controlled at 420 and 250 °C or 620 and 450 °C, respectively. When the temperature condition is 420/250 °C, the maximum $x$(Te) of the obtained crystals was around 0.25 even when the nominal composition of Te was 50% for samples with a size of 100 μm or more. When the temperature condition is 620/450 °C, the crystals with $x$(Te) exceeding 0.25 up to $x$(Te) ≈ 0.55 were obtained.

The $x$(Te) values were determined by the single-crystal XRD measurements. The XRD measurements have been performed by using a Rigaku XtaLAB P200 diffractometer with Mo-Kα radiation ($\lambda = 0.71073$ Å) at room temperature. The structures were solved by ShelXT[45] and refined by ShelXL[46] with Olex2[47] as a graphical user interface. The X-ray structural analysis was performed for crystals with typical size of ~50 μm to obtain $x$(Te) and lattice parameters. For larger samples used in the resistivity measurements at ambient pressure (Fig. 1d), we determined the $c$-axis length from the XRD measurements and the $x$(Te) values were calculated from the linear relationship in Fig. 1a.

**Low-temperature XRD measurements**. Synchrotron XRD measurements have been performed at beam line BL-8A in Photon Factory, KEK (High-Energy Accelerator Research Organization), Japan. The wave length of synchrotron radiation was $\lambda = 0.997$ Å, which was calibrated with $CeO_2$. For $x$(Te) ≈ 0.07 and 0.20, the samples were cooled by a helium gas-stream cooling method with the lowest temperature of ~40 K. For $x$(Te) ≈ 0.37 and 0.45, the samples were cooled by a He-refrigerator down to ~12 K.

**High-pressure measurements**. High-pressure resistivity measurements were performed in the Institute for Solid State Physics, University of Tokyo, with a constant-loading type CAC. The constant-loading type CAC can produce hydrostatic pressure and maintain a nearly constant pressure over the whole temperature range from 4.2 to 300 K. The maximum pressure used in our measurements was 8 GPa. For all high-pressure resistivity measurements, we employed glycerol as the pressure-transmitting medium, and used the conventional four-terminal method with current applied within the $ab$ plane. For samples denoted as $x$(Te) ≈ 0.18, 0.21, and 0.50 in Figs. 3 and 4, the $x$(Te) values are determined from the $c$-axis measured by the XRD. The $x$(Te) values of other $FeSe_{1-x}Te_x$ samples used in high-pressure measurements were determined from $T_s$ at ambient pressure by using the relationship between $T_s$ and $x$(Te) (the black line in Fig. 2a).

## Data availability

The data supporting the findings of this study are available within the paper. Any additional data connected to the study are available from the corresponding author upon reasonable request.

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

## Acknowledgements

The authors would like to thank H. Ikeda for helpful discussion and S. Nagasaki for technical assistance of high-pressure experiments. This work was partially carried out by the joint research in the Institute for Solid State Physics, University of Tokyo. The synchrotron X-ray study was performed with the approval of the Photon Factory Program Advisory Committee (No. 2017S2-001). This work was supported by Grants-in-Aid for Scientific Research (KAKENHI Grant Nos JP18K13492, JP18H05227, JP19H00649, JP18H01853, JP18KK0375, JP18J11320, JP19K22123, JP19H00648, JP20H02600, and JP20K21139), Grant-in-Aid for Scientific Research on Innovative Areas "Quantum Liquid Crystals" (KAKENHI Grant No. JP19H05824) from Japan Society for the Promotion of Science, and CREST (No. JPMJCR19T5) from Japan Science and Technology.

## Author contributions

K.Matsuura, M.Q., M.S., Y.S., and M.O. prepared single crystals. K.Mukasa, M.S., Y.S., K.I., Y.O., Y.M., K.H., and R.K. performed X-ray measurements. K.Mukasa, K.Matsuura, J.G., and Y.U. performed high-pressure measurements. K.Mukasa, K.Matsuura, Y.S., and T.S. analyzed the data. Y.U. and T.S. supervised the project. K.Mukasa and T.S. wrote the manuscript with inputs from all authors.

## Competing interests

The authors declare no competing interests.
