## [Peer Review File · Nature Communications]

REVIEWER COMMENTS

Reviewer #1 (Remarks to the Author):

This is a very well done and systematic study of the phase diagram of high quality $\text{FeSe}_{1-x}\text{Te}_x$. The authors used both chemical and hydrostatic pressure to determine if the nematic fluctuations can lead to higher superconducting T_c , which is a unique combination to explore these properties but I have a few concerns which won't necessarily change the substance of the manuscript but may give the authors insights on a future study.

1. Will high pressure XRD give the authors more insight on the crystal structure at specific contents given in Fig. 3? I'm curious if comparing the bulk modulus, and lattice parameters under pressure of FeSeTe vs. FeSSe will answer some questions risen in the discussion. It would also be nice to study the tetragonal vs. orthorhombic transition as a function of pressure at the different x contents.
2. For the non-monotonic behavior mentioned in Fig. 2, can you clarify the intrinsic origin of the T_c above $x(\text{Te}) = 0.30$? It's not clear that you refer to it in the discussion section.
3. For the T_m (magnetic transition temperature), is this a Curie or Néel temperature? From the looks of your data it seems like a ferromagnetic transition temperature, but you may want to clarify.
4. At what pressure do the authors think they would expect similar trends for low contents (0.04-0.21) versus high contents (0.38-0.50)? The SC transition temperature dies off at these higher contents but does not at the lower contents up to the max pressure of 8 GPa. I'm curious if you looked into higher pressures above 8 GPa by using a diamond anvil cell?
5. In Figure 3, I see that there was no manuscript text related to the strange shapes of the $R(T)$ curves at lower content, especially Fig. 3a around 2-5 GPa, are there any comments why this is? Is it related to potential defects? I'd expect that in the other samples as well but the data looks much more cleaner.

Overall, it is very nice result and I think this manuscript is suitable for publication in Nature Communications.

Reviewer #2 (Remarks to the Author):

Mukasa et al report high-pressure measurements on superconducting $\text{FeSe}_{1-x}\text{Te}_x$. The study is of fundamental interest as superconductivity in FeSe and the sulphur and tellurium substitution series is not yet understood. One important question is the relevance of nematic order and magnetic order as well as the relevance of quantum criticality of the two for superconductivity which the authors attempt to answer in their study. However, I have reservations both with the data analysis as well as the interpretation of the results and thus suggest a revision of the manuscript.

- 1) The authors argue that "the enhanced superconductivity correlates with the suppression of the nematic phase". The authors base this interpretation on the observation of a maximum superconducting transition temperature T_c in proximity to the critical pressure P_n of the nematic order. However, the maximum in T_c is at higher pressure compared to P_n in all their samples. Nematic fluctuations are largest at P_n . Thus, one would expect T_c to have the maximum at P_n if nematicity was

to mediate superconductivity. In fact, the phase diagrams could also be interpreted as a competition with superconductivity suppressed on approach of and inside the nematic phase.

2) The authors argue that "The magnetic phase in FeSe $1-x$ Te x moves quickly to the higher pressure side with x (Te)". The experimental evidence for this analysis is very weak. The SDW phase is observed for $x=0.04$ between 2 and 5 GPa and for $x=0.06$ between 1 and 5 GPa (Fig4a and b). Whilst the data for $x=0.10$ require further scrutiny before taking them into account (point 4). Taking this together with the data on sulphur doped FeSe presented in Fig4i one could come to the opposite conclusion that the magnetic phase is shifting to lower pressure for tellurium doped FeSe and that the SDW phase is determined by the pnictogen height. Even if the $x=0.10$ data are used, one could come to this conclusion for the SDW phase observed at 1 to 2 GPa.

3) The discussion of the relevance of SDW order for the superconductivity is very confusing (lines 182-188). It is not clear what the authors conclude. I have the impression that they favour a picture in which the superconductivity is driven by the nematic quantum criticality and not related to the magnetism. However, I urge the authors to discuss scenarios like superconductivity being suppressed by magnetism (as there is a dip in T_c in the SDW phase) and superconductivity arising from the SDW quantum criticality (T_c is maximum at critical pressure of SDW). Some other studies (which the authors omit to cite) have come to different conclusions e.g. [1,2].

4) The data analysis requires more scrutiny. The authors extract the nematic and SDW transition temperatures from anomalies in the resistivity. The signatures are difficult to see in some of the curves of Fig 3. In particular, the authors need to present evidence for the SDW transition for $x=0.10$ which is central for their interpretation (see point 2). In addition, the authors should discuss why the characteristics of the signature changes and yet are associated with the same phase transition, e.g. for $x=0.04$, the increase at 2 GPa and the decrease at 3 GPa are both assigned to the SDW phase.

5) I would also suggest the authors to relate their work to earlier pressure studies of FeSe with Te substitution [3]

In summary, I recommend extensive revision of the manuscript before considering it for publication in Nature Communications.

[1] Licciardello, S. et al. Electrical resistivity across a nematic quantum critical point. Nature 567, 213–217 (2019).at <<https://doi.org/10.1038/s41586-019-0923-y>>

[2] Reiss, P. et al. Quenched nematic criticality and two superconducting domes in an iron-based superconductor. Nature Physics 16, 89–94 (2020).at <<https://doi.org/10.1038/s41567-019-0694-2>>

[3] Panfilov, A. S. et al. Interrelation of superconductivity and magnetism in FeSe $1-x$ Te x compounds. Pressure effects. Low Temperature Physics 40, 615–620 (2014).(DOI:10.1063/1.4890990)

Reply to Reviewer #1

This is a very well done and systematic study of the phase diagram of high quality $\text{FeSe}_{1-x}\text{Te}_x$. The authors used both chemical and hydrostatic pressure to determine if the nematic fluctuations can lead to higher superconducting T_c , which is a unique combination to explore these properties but I have a few concerns which won't necessarily change the substance of the manuscript but may give the authors insights on a future study.

We thank Reviewer #1 for the very positive evaluation of our study. We respond to the comments one by one.

1. Will high pressure XRD give the authors more insight on the crystal structure at specific contents given in Fig. 3? I'm curious if comparing the bulk modulus, and lattice parameters under pressure of FeSeTe vs. FeSSe will answer some questions risen in the discussion. It would also be nice to study the tetragonal vs. orthorhombic transition as a function of pressure at the different x contents.

We thank for the reviewer for the helpful suggestion for the lattice parameters studies. As mentioned in the original manuscript, the Te and S substitutions correspond to negative and positive chemical pressure effects, respectively, and thus we agree that the crystal structure parameters are an important factor to discuss the difference between these two systems. We have analyzed the crystal structure as a function of Te content and its temperature dependence as shown in Fig. 1, which confirmed that the same nematic transition from tetragonal to orthorhombic structure occurs up to $x(\text{Te}) \sim 0.5$. The anomalies of the resistivity curves under pressure in Fig. 3 are very similar to those found in FeSe [21] and $\text{Fe}(\text{Se},\text{S})$ [22], thus we believe that the phase diagrams which is the main topic of the present paper have been obtained in a convincing method. The XRD measurements under pressure will provide additional information, as suggested by the reviewer, but require a completely different methodology from the present measurements (in which we use a constant-loading cubic anvil cell that can maintain nearly constant pressure with temperature), which we believe deserve further and separate studies.

2. For the non-monotonic behavior mentioned in Fig. 2, can you clarify the intrinsic origin of the T_c above $x(\text{Te}) = 0.30$? It's not clear that you refer to it in the discussion section.

The phase diagram implies that the increase of T_c above 0.30 is closely related to the suppression of T_s , which can be also seen in the pressure phase diagrams with the superconducting dome near the nematic end point (Fig. 4d-g). The nematic quantum fluctuations are expected to be enhanced near the nematic end point, and therefore, these phase diagrams indicate an intimate link between the T_c increase and nematic fluctuations. To make this point clearer, we have added discussion in the revised manuscript.

3. For the T_m (magnetic transition temperature), is this a Curie or Néel temperature? From the looks of your data it seems like a ferromagnetic transition temperature, but you may want to clarify.

From the high-pressure studies in FeSe, it has been found that the pressure-induced phase is an antiferromagnetic phase with stripe spin arrangement and orthorhombic structure, which is similar to those found in many other iron-based superconductors. Since the pressure-induced magnetic phase in the present Fe(Se,Te) is continuous from that in FeSe, it is natural to assign T_m as a Neel temperature. We have added explanations for this in the revised manuscript.

4. At what pressure do the authors think they would expect similar trends for low contents (0.04-0.21) versus high contents (0.38-0.50)? The SC transition temperature dies off at these higher contents but does not at the lower contents up to the max pressure of 8 GPa. I'm curious if you looked into higher pressures above 8 GPa by using a diamond anvil cell?

The maximum pressure we use for the present study is 8 GPa, because we focus on the cubic anvil cell measurements to compare with the results of Fe(Se,S) in Ref. [22] which have been obtained in the same condition. In the previous high-pressure studies of FeSe, the crystal structure changes at higher pressures: In a diamond anvil cell (DAC) studies for polycrystals [S. Medvedev *et al.*, Nat. Mater. **8**, 630 (2010)], the structure changes to hexagonal and the

resistivity exhibits semiconducting and non-superconducting behaviors above 30 GPa, but more recent single-crystal studies using a self-clamped cubic anvil cell [21] show that only 12-GPa pressure changes metallic to semiconducting resistivity behavior. This difference indicates that the hydrostatic pressure condition is important. The higher pressure measurements above 8 GPa in Fe(Se,Te) require a different setup as suggested by the reviewer, which will be a focus of future studies.

5. In Figure 3, I see that there was no manuscript text related to the strange shapes of the R(T) curves at lower content, especially Fig. 3a around 2-5 GPa, are there any comments why this is? Is it related to potential defects? I'd expect that in the other samples as well but the data looks much more cleaner.

The anomalies in the resistivity curves near T_m for $x=0.04$ (Fig. 3a) are very similar to those in FeSe, which have been thoroughly discussed in Refs. [21] and [T. Terashima *et al.*, Phys. Rev. B **93**, 180503(R) (2016)]. We have added discussion in the revised manuscript. When the antiferromagnetic transition occurs, the carrier number and the scattering rate both decrease, which results in a competition between the increase and decrease in the resistivity. Thus in some case such as for 2 GPa, resistivity jumps up below T_m , but in some other case such as for 3 GPa, resistivity drops below T_m . We use the same methodology as Refs. [21] and [22] to identify T_m by the peak or dip in the dp/dT in the present study.

Overall, it is very nice result and I think this manuscript is suitable for publication in Nature Communications.

We thank the reviewer for the recommendation of publication in Nature Communications.

=====

Reply to Reviewer #2

Mukasa et al report high-pressure measurements on superconducting FeSe_{1-x}Te_x. The study is of fundamental interest as superconductivity in FeSe

and the sulphur and tellurium substitution series is not yet understood. One important question is the relevance of nematic order and magnetic order as well as the relevance of quantum criticality of the two for superconductivity which the authors attempt to answer in their study. However, I have reservations both with the data analysis as well as the interpretation of the results and thus suggest a revision of the manuscript.

We thank Reviewer #2 for the in-depth reading of our manuscript. We find the reviewer's comments valuable. As suggested by the reviewer, we made extensive revisions which we believe address the reviewer's concerns.

1) The authors argue that "the enhanced superconductivity correlates with the suppression of the nematic phase". The authors base this interpretation on the observation of a maximum superconducting transition temperature T_c in proximity to the critical pressure P_n of the nematic order. However, the maximum in T_c is at higher pressure compared to P_n in all their samples. Nematic fluctuations are largest at P_n . Thus, one would expect T_c to have the maximum at P_n if nematicity was to mediate superconductivity. In fact, the phase diagrams could also be interpreted as a competition with superconductivity suppressed on approach of and inside the nematic phase.

We thank the reviewer for the valid concerns. Indeed, the maximum transition temperature is attained at somewhat different point from the extrapolated nematic end point. First of all, the exact position of the critical pressure of nematic order is difficult to pin down, because the superconductivity masks the critical point. Second, at the quantum critical point, the fluctuations are largest and thus the pairing interactions are expected to be largest, but at the same time the effect of quasiparticle damping may also become significant. In the case of ferromagnetic superconductors, for example, some theory suggests that the transition temperature of the ferromagnetic spin-fluctuation based unconventional superconductivity has a dip just at the critical point but becomes highest not far from the critical point [see, e.g., D. Fay and J. Appel, Phys. Rev. B **22**, 3173 (1980)]. It is thus not clear whether T_c should be highest exactly at the nematic critical point or not, but our phase diagrams show that the superconducting dome has a broad peak not far from the nematic end point. Third, most importantly, near the superconducting dome, we have no magnetic

order nearby and the closest order is the nematic order. As stated in the original manuscript, in the ambient pressure phase diagram in Fig. 2, the increasing trend of T_c is found above $x(\text{Te})\sim 0.3$, and near the end point $x(\text{Te})\sim 0.5$, T_c reaches close to the reported maximum value (14 K for $x(\text{Te})\sim 0.6$) in this system. It has also been reported from the NMR measurements that no significant antiferromagnetic fluctuations are present for $x(\text{Te})\sim 0.58$. These results clearly indicate that the enhanced superconductivity is intimately related to the end point of nematic order. Finally, as for the possible competition between nematicity and superconductivity, such a competition is expected even if the fluctuations of competing order promote superconductivity. This can be understood by the fact that once the long-range order is established below T_s , the nematic fluctuations are suppressed. Similar suppressions of superconductivity inside the antiferromagnetic phase have been observed in many cases including the pressure-induced antiferromagnetic phase in FeSe-based superconductors. To clarify these points, we have added extended discussion in the revised manuscript.

2) The authors argue that “The magnetic phase in FeSe $1-x$ Te x moves quickly to the higher pressure side with $x(\text{Te})$ ”. The experimental evidence for this analysis is very weak. The SDW phase is observed for $x=0.04$ between 2 and 5 GPa and for $x=0.06$ between 1 and 5 GPa (Fig4a and b). Whilst the data for $x=0.10$ require further scrutiny before taking them into account (point 4). Taking this together with the data on sulphur doped FeSe presented in Fig4i one could come to the opposite conclusion that the magnetic phase is shifting to lower pressure for tellurium doped FeSe and that the SDW phase is determined by the pnictogen height. Even if the $x=0.10$ data are used, one could come to this conclusion for the SDW phase observed at 1 to 2 GPa.

We agree to the reviewer that for $x=0.10$, we also have the magnetic phase at low pressure (1 GPa) as shown in Fig. 4c, in addition to the high-pressure side. We have focused on the high-pressure side which goes away for $x=0.14$ which is smaller than the S-substitution case, but after reading the reviewer’s comment we have realized that the statement may have been misleading. Thus we have revised the corresponding discussion to a more accurate description of the results. We thank the reviewer for pointing this out.

3) The discussion of the relevance of SDW order for the superconductivity is very confusing (lines 182-188). It is not clear what the authors conclude. I have the impression that they favour a picture in which the superconductivity is driven by the nematic quantum criticality and not related to the magnetism. However, I urge the authors to discuss scenarios like superconductivity being suppressed by magnetism (as there is a dip in T_c in the SDW phase) and superconductivity arising from the SDW quantum criticality (T_c is maximum at critical pressure of SDW). Some other studies (which the authors omit to cite) have come to different conclusions e.g. [1,2].

What we conclude here is that in the studies of Fe(Se,S) series, in which there is no enhancement of T_c at the nematic end point, there is no clear evidence that the nematic fluctuations promote superconductivity. In lines 182-186 in the original manuscript we tried to point out that nematic fluctuations in addition to spin fluctuations may be present near the end point of pressure-induced antiferromagnetism, because this state accompanies the orthorhombicity as in the iron-pnictide case. However, this cannot be taken as evidence for nematic-fluctuation driven superconductivity, which was not clear, and we have removed the corresponding sentences. We have extensively revised the introduction and discussion paragraphs to make these points clearer, with the relevant citations suggested by the reviewer. In particular, we added the possibility of quenched nematic criticality by the strong coupling to the lattice in Fe(Se,S) discussed in the suggested reference [2] which has been missed in the original submission.

4) The data analysis requires more scrutiny. The authors extract the nematic and SDW transition temperatures from anomalies in the resistivity. The signatures are difficult to see in some of the curves of Fig 3. In particular, the authors need to present evidence for the SDW transition for $x=0.10$ which is central for their interpretation (see point 2). In addition, the authors should discuss why the characteristics of the signature changes and yet are associated with the same phase transition, e.g. for $x=0.04$, the increase at 2 GPa and the decrease at 3 GPa are both assigned to the SDW phase.

We use the same methodology as Refs. [21] and [22] to identify T_m by the peak or dip in the $d\rho/dT$ in the present study. For $x=0.10$ we see these anomalies at

the points showing by the arrows. The anomalies in the resistivity curves near T_m for $x=0.04$ (Fig. 3a) are very similar to those in FeSe, which have been thoroughly discussed in Ref. [21]. When the antiferromagnetic transition occurs, the carrier number and the scattering rate both decrease, which results in a competition between the increase and decrease in the resistivity. Thus in some case such as for 2 GPa, resistivity jumps up below T_m , but in some other case such as for 3 GPa, resistivity drops below T_m . These are assigned as the same antiferromagnetic (SDW) phase because the transition temperature shows a systematic change. In the case of FeSe, the field dependence of these anomalies has been measured and sometimes the jump changes to a drop in the resistivity as a function of field (see Fig. 4b of [21]), but the transition temperature remains unchanged, consistent with the antiferromagnetic transition. We have added discussion on this point in the revised manuscript.

5) I would also suggest the authors to relate their work to earlier pressure studies of FeSe with Te substitution [3]

We thank the reviewer for bringing this relevant citation to our attention. We cite this with some discussion that the positive dT_c/dP reported there up to $x(\text{Te})\sim 0.75$ is consistent with our results.

In summary, I recommend extensive revision of the manuscript before considering it for publication in Nature Communications.

[1] Licciardello, S. et al. Electrical resistivity across a nematic quantum critical point. *Nature* 567, 213–217 (2019).at <https://doi.org/10.1038/s41586-019-0923-y>

[2] Reiss, P. et al. Quenched nematic criticality and two superconducting domes in an iron-based superconductor. *Nature Physics* 16, 89–94 (2020).at <https://doi.org/10.1038/s41567-019-0694-2>

[3] Panfilov, A. S. et al. Interrelation of superconductivity and magnetism in FeSe $_{1-x}$ Te $_x$ compounds. Pressure effects. *Low Temperature Physics* 40, 615–620 (2014).(DOI:10.1063/1.4890990)

We believe we have addressed all the issues raised by the reviewer, and with extensive revisions, our paper is much improved by the valuable comments from the reviewer.

REVIEWER COMMENTS

Reviewer #1 (Remarks to the Author):

I thank the authors for addressing my comments so thoroughly. I appreciate the effort of this work, and think it is suitable for publication in Nature Communications at this stage without further delay.

Best,

Dante J. O'Hara, Ph.D.

Reviewer #2 (Remarks to the Author):

The manuscript presents important measurements that map the superconducting, nematic, and magnetic phases. The main claim of the paper is that nematic fluctuations promote superconductivity in FeSeTe. The revisions address some of my comments and the main claim is well supported by evidence although should be discussed more critically. In addition, some of my previous points remain unresolved in my opinion. Hence, I urge the authors to consider the points below before the manuscript is considered for publication.

1) I accept the arguments of the authors that superconductivity mediated by nematicity can have a maximum away from the critical pressure. This addition to the manuscript is valuable for readers. However, the authors do not discuss the possibility of competition between superconductivity and nematicity. Whilst they make some arguments in their rebuttal, this is not included in the manuscript. In addition, the arguments in the rebuttal cannot rule out that the leading effect between superconductivity and nematicity is competition. Hence, I suggest a more balanced discussion of this main claim of the paper.

2) Likewise, there is no evidence in the manuscript or the discussion of literature that rules out a competition between superconductivity and magnetism. As I suggested before, the dip in T_c in the magnetic phase could also be interpreted that magnetism is suppressing superconductivity. In fact, this resonates with the arguments of the authors that magnetism is not promoting superconductivity. Hence, I suggest the authors to consider this possibility.

3) In my first review point 2 and 4, I have questioned the analysis and conclusion of the pressure range where magnetic order exists. No new evidence, analysis, or revision of the conclusion has been done on these points. Specifically, the maintained claim that the position of pressure range of magnetic order is not linked to the pnictogen height hinges on the data for $x=0.1$ Te. However, this data is not convincing. The signatures for magnetism in $x=0.1$ Te at 5, 6, and 7 GPa are small and of different shape to most of the other transitions to the magnetic state. In fact, these signatures are smaller than the step in $x=0.21$ and 4GPa at 45K which is probably a measurement artifact. Given the weak evidence for AFM at $x=0.1$ and 5-7GPa, I suggest the authors to tone down their claim on the relation between the magnetic phase and pnictogen height. In addition, I suggest the authors show the derivatives of the resistivity used for the analysis to allow the reader to inspect the evidence for this claim.

4) The language of the manuscript needs revision, e.g.

Line 15: „have been focused“ probably should be „have been suggested“

Line 21: „x(S)“ is not defined

Line 21: referring to „above fundamental question“ whilst there is no clear question above.

Line 35: „has been believed“ suggests that this is no longer the case. Probably the authors don't mean that.

Line 39 and more: „glue“ is rather colloquial

Line 41: „but“ should probably rather be „and“

Line 43: „the both ends“ should probably be „both ends“

Line 57: „nonmagnetic“ should probably be „magnetic“ Otherwise, it would contradict the main claim of the paper.

Line 80: „ that the phase separation“ should probably be „that phase separation“

Line 83: „a few contents“ could be replaced by „a few compositions“

Line 90: „that the Vegard's law“ should probably be „that Vegard's law“

Line 92: „behaviours“ -> „behaviour“

Line 94: „speared“ -> „spread“

Line 124: „stipe-type“ -> „stripe-type“

Line 127: „increase in scattering rate“ should probably be „decrease in scattering rate“

Line 167: “near the both ends” -> “near both ends”

Figure 1 caption needs to define “split width”. In fact, it is not clear how this differs from δ

Reply to Reviewer #1

I thank the authors for addressing my comments so thoroughly. I appreciate the effort of this work, and think it is suitable for publication in Nature Communications at this stage without further delay.

We thank Reviewer #1 for the full endorsement.

=====

Reply to Reviewer #2

The manuscript presents important measurements that map the superconducting, nematic, and magnetic phases. The main claim of the paper is that nematic fluctuations promote superconductivity in FeSeTe. The revisions address some of my comments and the main claim is well supported by evidence although should be discussed more critically. In addition, some of my previous points remain unresolved in my opinion. Hence, I urge the authors to consider the points below before the manuscript is considered for publication.

We thank Reviewer #2 for the second in-depth review of our manuscript. We take the reviewer's comments seriously and now incorporated most of them, which we found helpful to improve our paper.

1) I accept the arguments of the authors that superconductivity mediated by nematicity can have a maximum away from the critical pressure. This addition to the manuscript is valuable for readers. However, the authors do not discuss the possibility of competition between superconductivity and nematicity. Whilst they make some arguments in their rebuttal, this is not included in the manuscript. In addition, the arguments in the rebuttal cannot rule out that the leading effect between superconductivity and nematicity is competition. Hence, I suggest a more balanced discussion of this main claim of the paper.

2) Likewise, there is no evidence in the manuscript or the discussion of literature that rules out a competition between superconductivity and magnetism. As I suggested before, the dip in T_c in the magnetic phase could also be interpreted that magnetism is suppressing superconductivity. In fact, this resonates with the

arguments of the authors that magnetism is not promoting superconductivity. Hence, I suggest the authors to consider this possibility.

We appreciate the reviewer's suggestion to make the discussion more balanced. As pointed out by the reviewer, the competition between any long-range orders and superconductivity may result in the suppression of T_c inside the ordered phase. Indeed, the T_c suppression found inside the pressure-induced magnetic phase in FeSe accompanies a kink behavior of $T_c(P)$ at the crossing point with $T_m(P)$, which can be explained by the competition mechanism between magnetism and superconductivity. We have incorporated this reviewer's viewpoint in the revised manuscript.

The competition between nematicity and superconductivity can also explain the opposite trends between T_c and T_s as functions of $x(\text{Te})$ and pressure inside the nematic phase. However, this competition alone cannot explain the superconducting dome we observed centered outside the nematic phase. Moreover, it has been established that the conventional phonon pairing mechanism cannot reproduce the high T_c values in iron-based superconductors and thus we need some unconventional mechanism to explain the high- T_c dome outside the nematic phase. It is then important to consider existing bosonic fluctuations that lead to electron pairing and how they change as a function of nonthermal parameters in the phase diagrams. Our point is that the superconducting domes are found near the nematic end point, not close to the magnetic phase, which implies close relationship between nematic fluctuations and enhanced superconductivity. As we mentioned in the previous reply, this scenario does not contradict the competition effect inside the ordered phase, because nematic fluctuations are expected to be suppressed with the development of nematic order. We have added a paragraph in the discussion section on this point which indeed improve our manuscript.

3) In my first review point 2 and 4, I have questioned the analysis and conclusion of the pressure range where magnetic order exists. No new evidence, analysis, or revision of the conclusion has been done on these points. Specifically, the maintained claim that the position of pressure range of magnetic order is not linked to the pnictogen height hinges on the data for $x=0.1$ Te. However, this data is not convincing. The signatures for magnetism in $x=0.1$ Te at 5, 6, and 7

GPa are small and of different shape to most of the other transitions to the magnetic state. In fact, these signatures are smaller than the step in $x=0.21$ and 4GPa at 45K which is probably a measurement artifact. Given the weak evidence for AFM at $x=0.1$ and 5-7GPa, I suggest the authors to tone down their claim on the relation between the magnetic phase and pnictogen height. In addition, I suggest the authors show the derivatives of the resistivity used for the analysis to allow the reader to inspect the evidence for this claim.

As suggested by the reviewer, we now show the data for temperature derivatives of resistivity from which we identify the magnetic transition temperature T_m (see Supplementary Information). As pointed out by the reviewer, the step found in $x=0.21$ is likely a measurement artifact, but we can see clear anomalies of dp/dT in $x=0.10$ at 5, 6, 7 GPa, which resemble the magnetic anomalies found in lower compositions. However, we agree to the reviewer that the origin of the pressure-induced magnetic phase may be difficult to discuss. Accordingly, we have removed the corresponding statement on the relation between the magnetic phase and pnictogen height.

4) The language of the manuscript needs revision, e.g.

Line 15: „have been focused“ probably should be „have been suggested“

Line 21: „ $x(S)$ “ is not defined

Line 21: referring to „above fundamental question“ whilst there is no clear question above.

Line 35: „has been believed“ suggests that this is no longer the case. Probably the authors don't mean that.

Line 39 and more: „glue“ is rather colloquial

Line 41: „but“ should probably rather be „and“

Line 43: „the both ends“ should probably be „both ends“

Line 57: „nonmagnetic“ should probably be „magnetic“ Otherwise, it would contradict the main claim of the paper.

Line 80: „ that the phase separation“ should probably be „that phase separation“

Line 83: „a few contents“ could be replaced by „a few compositions“

Line 90: „that the Vegard's law“ should probably be „that Vegard's law“

Line 92: „behaviours“ -> „behaviour“

Line 94: „speared“ -> „spread“

Line 124: „stipe-type“ -> „stripe-type“

Line 127: „increase in scattering rate“ should probably be „decrease in scattering rate“

Line 167: “near the both ends” -> “near both ends”

Figure 1 caption needs to define “split width”. In fact, it is not clear how this differs from δ

We truly thank the reviewer’s suggestions for these corrections. We have incorporated them as indicated by blue colors in the text. As for the caption for the X-ray diffraction data in Fig.1f, the orthorhombicity δ can be determined by $(hk0)$ Bragg peaks, but for some data we can only evaluate the split width of (hkl) peaks with nonzero l , from which the orthorhombicity cannot be determined.

We believe we have addressed all the issues raised by the reviewer, and with these revisions, we hope that the reviewer finds our paper suitable for publication in *Nature Communications*.